# *It's made a really hard situation even more difficult*: The impact of COVID-19 on families of children with chronic illness

**Jordana McLoone**[1,2]*, **Claire E. Wakefield**[1,2], **Glenn M. Marshall**[3], **Kristine Pierce**[1], **Adam Jaffe**[1,4], **Ann Bye**[1], **Sean E. Kennedy**[1,5], **Donna Drew**[3], **Raghu Lingam**[1]

1 School of Clinical Medicine, UNSW Medicine and Health, UNSW Sydney, Sydney, New South Wales, Australia, 2 Behavioural Sciences Unit, Kids Cancer Centre, Sydney Children's Hospital, Sydney, New South Wales, Australia, 3 Kids Cancer Centre, Sydney Children's Hospital, Sydney, New South Wales, Australia, 4 Respiratory Department, Sydney Children's Hospital, Randwick, New South Wales, Australia, 5 Nephrology Department, Sydney Children's Hospital, Randwick, New South Wales, Australia

* J.McLoone@unsw.edu.au

**Data Availability Statement:** Data cannot be shared publicly in order to uphold the confidentiality of participants who have shared confidential medical information pertaining to

## Abstract

### Objective

For over two years, the global COVID-19 pandemic has forced major transformations on health, social, and educational systems, with concomitant impacts on mental health. This study aimed to understand the unique and additional challenges faced by children with chronic illness and their families during the COVID-19 era.

### Method

Parents of children receiving treatment for a chronic illness within the neurology, cancer, renal and respiratory clinics of Sydney Children's Hospital were invited to participate. We used qualitative methodology, including a semi-structured interview guide, verbatim transcription, and thematic analysis supported by QSR NVivo.

### Results

Thirteen parents of children receiving tertiary-level care, for nine chronic illnesses, participated. Parents reported intense fears relating to their ill child's additional vulnerabilities, which included their risk of developing severe COVID-19 disease and the potential impact of COVID-19-related disruptions to accessing clinical care, medications, allied health support and daily care protocols should their parent contract COVID-19. Parents perceived telehealth as a highly convenient and preferred method for ongoing management of less complex healthcare needs. Parents reported that the accrual of additional stressors and responsibilities during the pandemic, experienced in combination with restricted social interaction and reduced access to usual support networks was detrimental to their own mental health. Hospital-based visitation restrictions reduced emotional support, coping, and resilience for both parents and children and in some cases led to marital discord, sibling distress, and financial loss. Supportive factors included increased time spent together at home during

children. In addition, we are restricted by the relevant ethics committee to not allow access to this data unless expressly named in the approved application. All requests for access to data can be made to either Jordana Mcloone (J. Mcloone@unsw.edu.au) or to the ethics committee, quoting the study reference number: 2020/ETH02434. The Sydney Children's Hospital Network Human Research Ethics Committee (SCHN HREC) Email: SCHN-Ethics@health.nsw. gov.au Phone +612 9845 1253. Mail: 'Research Ethics' at the Sydney Children's Hospital Network, Locked Bag 4001, Westmead, 2145, NSW, Australia.

**Funding:** The author(s) received no specific funding for this work.

**Competing interests:** The authors have declared that no competing interests exist.

the pandemic and improved hygiene practices at school, which dramatically reduced the incidence of non-COVID-19-related communicable illnesses in chronically ill children.

## Discussion

For families caring for a chronically ill child, COVID-19 made a difficult situation harder. The pandemic has highlighted the need for targeted psychosocial intervention for vulnerable families, to mitigate current mental health burden and prevent chronic psychological distress.

## Introduction

Globally, COVID-19 has caused disruptions to the clinical management of chronic childhood illnesses, including reduced access to screening, diagnostic and therapeutic services, surgery, essential medications, and follow-up surveillance [1–4]. Arising in parallel with the medical challenges of the pandemic are immense changes to the social landscape. The lifestyle and needs of the child, including education and peer interactions, have been curbed by social isolation and school closures [5–8]. For children with a chronic illness, who already face a disproportionate level of psychosocial burden [9], it is expected that this confluence of factors may result in additional consequences compared to the general population [10].

Across childhood chronic illnesses, parents fill gaps in fragmented and uncoordinated healthcare systems, functioning as case managers, medical record keepers, and patient advocates. The pandemic required parents to rapidly navigate their child's new care pathways, as health systems became overwhelmed and healthcare delivery models pivoted to telehealth and virtual care [11, 12]. Simultaneously, workplaces and schools also transitioned to virtual sites, with little time for planning, piloting, or training. Up-skilling to new work platforms, procedures, and home-learning responsibilities, while caring for a chronically ill child, is likely to have placed parents under unprecedented levels of pressure. Not surprisingly, emerging evidence indicates that parents of children with underlying health conditions have been experiencing greater stress than parents of healthy children during the pandemic [13, 14].

Given the vital role families play in the health of children with chronic illness, a deeper understanding of their pandemic experiences is needed to ensure that evidence-based research guides decision-makers as they rapidly develop and deploy new services, and also re-imagine what a return to a new normal might look like for child healthcare services. This study therefore explored the impact of COVID-19 on families of children with chronic illness.

## Method

We used an explorative, qualitative methodology to obtain nuanced insights into the experiences of children with chronic illness and their families during the COVID-19 pandemic. We first conducted a narrative literature review of the emerging literature on COVID-19 and its impact on children with disability or illness. This review was guided by the Economic and Social Research Council framework [15] and we presented this to a multidisciplinary panel, including five pediatric specialists, a behavioral scientist, and a parent consumer. The expert panel reviewed the literature and discussed the potential issues relevant to children with chronic illness in the context of the pandemic and developed a semi-structured interview guide through an iterative process. The final interview guide explored families' experiences of

the COVID-19 pandemic and how their access to medical care, education, and social support had been impacted. The interview also explored the mental health impact of caring for a chronically ill child during a pandemic, as well as the mental health impacts for other family members, including the child with chronic illness and their sibling(s). At the conclusion of the interview we collected demographic items (e.g. age and sex of each family member).

First author JM (PhD), who has over a decade of qualitative research experience, interviewed families between November, 2020 and September, 2021. We audio-recorded interviews, which were professionally transcribed verbatim. We organized coded passages into themes and analysis was guided by the Braun and Clarke approach [16]. We used QSR NVivo 12 software to support data coding and analysis [17]. We used the COREQ [18] checklist to promote accurate reporting of qualitative studies. The Sydney Children's Hospital Network ethics board granted ethical approval (2020/ETH02027).

## Sample

We invited parents of children who a) were currently less than 16 years of age, b) had received treatment at Sydney Children's Hospital (SCH) within the past year, and c) were diagnosed with a chronic illness (a persistent illness requiring long term management), to participate in the study. SCH clinics including neurology, renal, respiratory and cancer, were asked to provide the contact details of families who met the study's eligibility criteria and were also purposively selected to provide a range of experiences, including recently diagnosed/long term patients, urban/rural, and severe/moderate disease. We sent families an invitation letter and consent form via mail and telephoned two weeks later to further explain the study and confirm whether or not a parent was interested in participating.

## COVID-19 context

The COVID-19 context in which these interviews were conducted was as follows. Sydney Children's Hospital is located in the Australian state of New South Wales (NSW). NSW has a population of over 8 million residents, yet between 1 January 2020 and 1 June 2021, reported only 5,587 cases of COVID-19. There were 54 deaths during this period, with 13 of these deaths occurring outside of aged care services [19]. For many months, there were zero to few cases of community transmission. Schools were "closed" for approximately six weeks, however the children of essential workers continued to be allowed to attend school during this period.

## Results

Thirty families were invited to participate. Of these, 13 mothers participated (43% response rate), see Table 1. Interviews were, on average, 42 minutes in length (range: 17–61 minutes). Five major themes were identified; COVID-19 specific concerns and impacts on medical care, mental health, support, and education. These themes are explored in detail below and illustrated in Fig 1.

### COVID-19 specific concerns

Parents reported that the pandemic had been a highly stressful period and described how, for their families, the pandemic stress was borne in addition to multiple other stressors. *"I mean things are already stressful in this house—a new baby and a chronically ill kid and you throw in COVID and job loss . . .things were intense a lot here."* (ID 09, child with end stage kidney failure).

Parents of chronically ill children reported general feelings of uncertainty and fear, as well as fears unique to their child's chronic health condition. *"I was terrified at the beginning of the*

**Table 1. Participant demographic characteristics.**

| Participant characteristic | N = 13 |
| --- | --- |
| Parent sex, n(%) | 13 female (100%) |
| Parent age, mean (range) | 42.3 years (range: 36–52 years) |
| Sex of child with chronic illness, n(%) | 61.5% male |
| Age of child with chronic illness, mean (range) | 8.5 years (range: 4–16 years) |
| Number of siblings of child with chronic illness, mean (range) | 1.8 siblings (range: 1–7) |
| Sibling age, mean (range) | 12.5 years (range: 0–24 years) |
| Chronic illness diagnosis* | Cancer |
| | Pre B Acute Lymphoblastic Leukemia |
| | Acute Lymphoblastic Leukemia |
| | Acute Lymphoblastic Leukemia |
| | Rhabdomyosarcoma |
| | Renal |
| | Nephrotic syndrome |
| | Posterior urethral valves |
| | End stage kidney failure |
| | Mid-aortic syndrome |
| | Neurology |
| | Epilepsy |
| | Epilepsy (neuronal migration disorder) |
| | Epilepsy (cortical dysplasia) |
| | Other |
| | Cerebral Palsy |
| | Severe asthma |
| | Bronchiectesis |
| | Dextrocardia situs inversus |

* Some children were diagnosed with multiple chronic illnesses and as such, more than 13 illnesses are reported.

pandemic. *I just thought every time [my son] gets a cold he ends up in the hospital and I just thought I don't wanna think about what will happen if he gets this!"* (ID 06, child with severe asthma). Parents' fears were typically intensified by extreme worries that the virus might have a greater impact on their chronically ill child, relative to children without chronic illness. *"I've watched her crash and go to ICU. You know what a virus can do."* (ID 10, child with epilepsy). Parents of chronically ill children often reported taking precautions and self-restricting movement beyond what the Australian public health guidelines recommended and continued to do so during periods when there were no, or very few (<10), community cases in the state. *"I didn't go to the shops until Monday [December]. I hadn't been out like from February."* (ID 09, child with end stage kidney failure).

Parents' fears centered on what might happen to their child if they became ill and could not support the daily medical needs of their child (see S1 Table for extended quotes). Parents worried that should a member of the family contract COVID-19, their child's chronic illness-related treatment would be postponed until the family completed their mandatory home-isolation period. *"That was my major anxiety last year, that one of us would get COVID and that would jeopardise his treatment. So that was my biggest concern."* (ID 02, child with rhabdomyosarcoma). Families also reported fears that medical supplies might be disrupted during the pandemic. *"Because we rely on so many things to keep our little girl alive, I was definitely scared that things like that would be impacted."* (ID 09, child with end stage kidney failure) and in

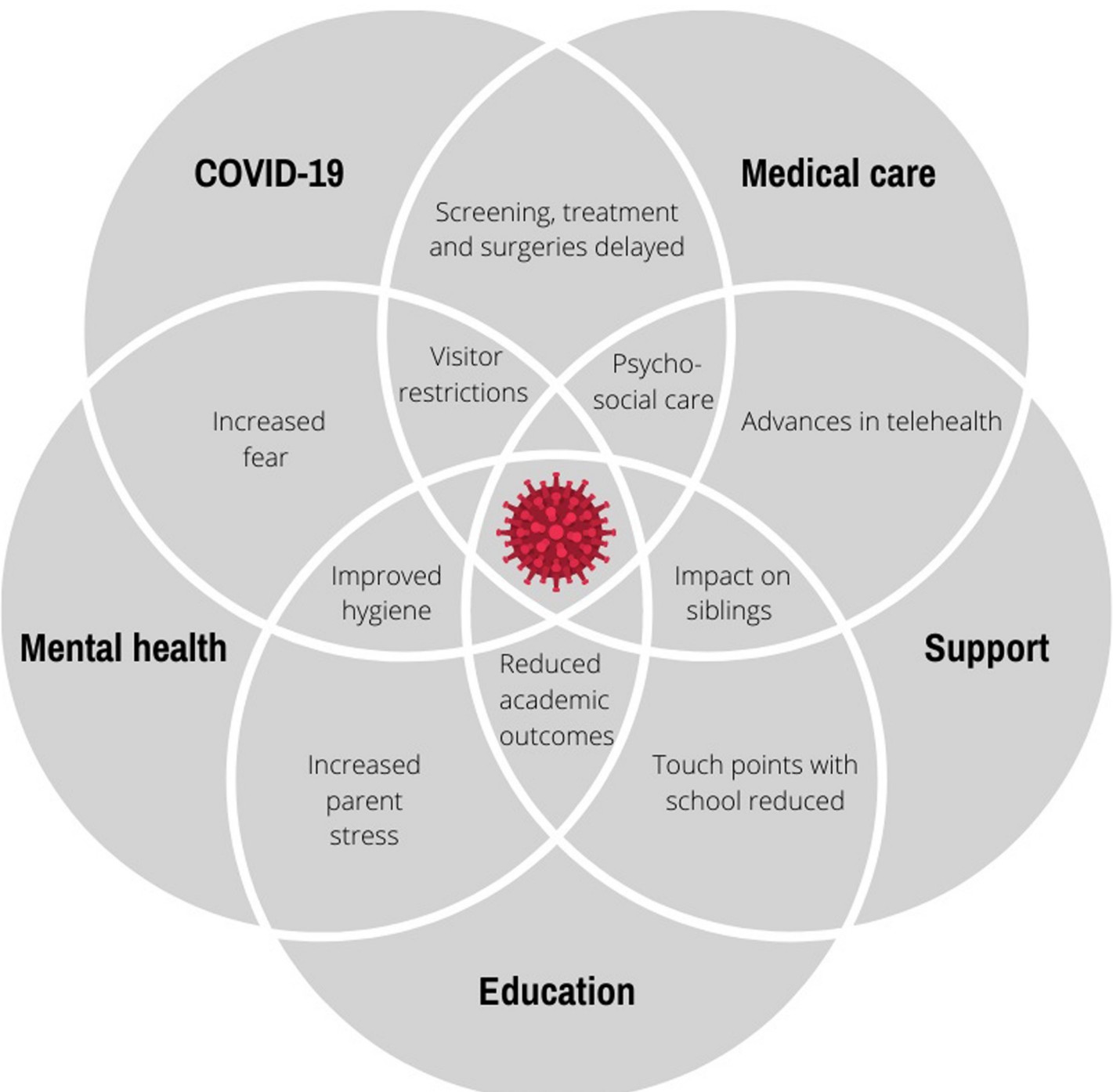

**Fig 1. Complexity diagram representing the inter-relatedness of factors impacting families of children with chronic illness during the COVID-19 pandemic.**

2021, when the Delta strain caused case numbers to surge, parents were fearful that there would not be sufficient hospital beds available for their chronically ill child. *"I'm concerned if they [can] proceed with surgery that the beds will all be filled with anti-vaxxers and cases of COVID."* (ID 13, child with epilepsy (cortical dysplasia))

In terms of the emerging COVID-19 vaccines, some parents were concerned that there may be insufficient trialing of the vaccine in children with rare diseases, with potentially unknown risks. *"I am also worried about vaccination because [my daughter] has a chronic illness. Like, is it going to be tested enough on people like that?"* (ID 09, child with end stage kidney failure).

## COVID-19 interruptions to medical care

Despite many challenges and disruptions, many families reported that their child's acute care continued uninterrupted, and that telehealth had supported ongoing virtual clinics and communication between families of children with chronic illness and hospital clinicians. Many families felt that communication with their team had not been disrupted. *"The hospital team's been really supportive through it all. Always really impressed—we've been able to get onto them and everyone's been really informative."* (ID 10, child with epilepsy)

In terms of the ongoing management of chronic illnesses, some parents reported that clinics had been cancelled, *"I think they cancelled all eye clinic reviews for us for nearly a year."* (ID 10, child with epilepsy). In addition, certain medications had not been available at the hospital pharmacy. *"I get a phone call from pharmacy and she goes, "Look, you can't take all the medication because you know, with COVID, we have a shortage."* (ID 12, child with Dextrocardia situs inversus). There were also considerable delays reported regarding critical scans and surgeries. *"He actually needed surgery again but because of COVID . . .it was delayed for a long time. He was getting quite unwell and we just had to wait."* (ID 08, child with mid-aortic syndrome)

Some families were burdened with the difficult decision of deciding whether or not to proceed with surgery, weighing the potential risks (exposure to COVID-19 within the hospital) and benefits (of surgery) for their child. Delays to surgery were often beyond the control of families and caused significant anxiety among parents, as they watched their child's health deteriorate. In addition, surgical delays often resulted in other developmental losses for young children, *"Huge worries about the future. [For] a 5 year old, a 6 months' delay [to surgery] is a huge portion of their life and it's such formative years in terms of their developmental opportunities."* (ID 13, child with epilepsy (cortical dysplasia)). For some children, accessing potentially life-altering surgery at centers of excellence remained uncertain due to the closure of state borders.

Clinicians used videoconferencing platforms to conduct virtual clinics, which were perceived as easy to use and highly convenient *"There are times [my daughter] might just not be feeling great, or we are worried because it's winter . . .so we just do telehealth."* (ID 09, child with end stage kidney failure). The convenience of virtual clinics also included reducing travel burden, eliminating travel sickness, and fitting around other family commitments, for example, the schedules of a newborn baby. *"It's nice not having to drive from Canberra to Sydney [3-hour drive], as I had a little baby."* (ID 09, child with end stage kidney failure). Telehealth was seen as a tool that would also greatly benefit rural and regional families *"I think for regionally based people telehealth is great."* (ID 07, child with posterior urethral valves) and enhance interstate collaborations. Telehealth also supported the continuation of allied health services throughout the pandemic and families reported that this was convenient in many ways. *"The use of technology was forced upon everyone pretty quickly, but it certainly has its benefits."* (ID 07, child with posterior urethral valves).

Families of children with chronic illness reported that their general practitioner's (GP) use of telehealth was telephone-based (video-conferencing was not available) and that this this was the preferred method for low-risk, or ongoing management, for example, *"If it's something little that can be done over the phone, like a script or something"* (ID 03, child with nephrotic syndrome). Avoiding the contagion risks associated with the GP waiting room was also noted as a

benefit of care provided via telehealth. *"I prefer not to take myself to a GP and risk catching something that I could give to [my chronically ill son] if I don't have to. So [I prefer] not taking my daughter there, or myself."* (ID 04, child with acute lymphoblastic leukemia). Also, for families whose child was wheelchair bound, or who were less mobile due to their child's chronic illness, telehealth with the GP was preferred. *"So it's made it so much easier, it's brilliant actually it's made life 100% better."* (ID 01, child with cerebral palsy)

Given the convenience of telehealth, many families hoped that it would remain available as an option after the pandemic was over and cited advances in providing care remotely as a silver lining of the pandemic. The increased availability of and access to care that telehealth offers rural and remote families was also discussed. *"If we lived in a regional area [telehealth] would make a huge difference to accessing treatment."* (ID 13, child with epilepsy (cortical dysplasia)).

## The impact of COVID-19 interruptions on mental health

Some families reported that they had been coping well during the pandemic. Protective factors included higher resilience among families with a chronically ill child relative to other families, learning about self-care during earlier waves of the pandemic, and utilizing virtual support groups. *"I think there is a certain amount of resilience and adaptability that you have when you've got a child with a chronic illness."* (ID 06, child with severe asthma). Many families reported that the ways in which the pandemic had forced them to slow down, limit extracurricular activities, and spend more time together as a family, had been beneficial. *"We were happy with just the four of us and our little family for a while."* (ID 03, child with nephrotic syndrome)

Parents recognized how important their own mental health was during this difficult time, as they needed to stay strong for their sick child. *"I've got an added responsibility to manage my feelings... I don't want to impact his situation [and make it] more difficult because I'm not managing how I feel."* (ID 13, child with epilepsy (cortical dysplasia)). Parents noted that they were having to manage their own mental health needs as they journeyed through their child's illness path,

> *"The main thing that I've struggled with is sitting with the current uncertainty and trying to hold hope but also prepare myself for the possibility that it's not treatable. I feel more despair and more hopelessness. Looking after [my son] I guess I've got an added responsibility to manage my feelings. I don't want to impact his situation and make it more difficult because I'm not managing how I feel."* (ID 13, child with epilepsy)

This was in addition to the needs of their chronically ill child *"It's a bit of a rollercoaster ride with her, still she's just sad."* (ID 10, child with epilepsy), and siblings' mental health too, *"he's ended up with a bit of ongoing anxiety since lockdown; which we are still dealing with now."* (ID 06, child with severe asthma).

Mothers recognized how much strain they had experienced to support others in the family, especially under circumstances that saw them also juggling multiple practical roles. Many mothers reported limited support for themselves, including not being able to see, or physically touch their friends during the most difficult times. *"When you're upset or something terrible happens you would hug your friends. ...I stopped that altogether."* (ID 11, child with acute lymphoblastic leukemia). Some mothers reported that it had been difficult to lose their supportive work environment during this time and missed the reprieve that their career experiences gave them in a face-to-face work environment.

While some parents reported that they were aware that excellent support for their child was available via their child's hospital psychologist and social worker, *"I've spoken to [the clinical*

*psychologist] a few times and she's amazing.*" (ID 05, child with pre B acute lymphoblastic leukemia), others reported that there were barriers to receiving mental health support during the pandemic, especially in the community setting. The pandemic had increased the need for services, at a time when services were limited, causing a backlog in private mental health services and long waitlists.

Many families were separated by pandemic-restrictions and the negative impact of this was consistently reported. To limit COVID-19 transmission risk, the hospital implemented a 'one-parent, no siblings' rule for most months from March, 2020 onwards. While parents understood the hospital's policy was aimed at protecting its patients and staff, they also felt this policy placed families at risk of mental health impacts as it limited the emotional support they could provide each other at critical times. Parents emphasized the importance of communicating well with each-other, especially during traumatic periods. Parents reported difficulty communicating their own emotional needs, as well as distressing and complex medical information to the second parent who was not allowed to enter the hospital.

Although parents were appreciative of the circumstance, they consistently emphasized the negative impact of siblings being unable to see each-other during hospital admissions. "*I'd say that probably had the biggest impact on the family.*" (ID 11, child with acute lymphoblastic leukemia). Mothers of young siblings (e.g. breast-feeding infants) also found the experience to be distressing. "*I'm breastfeeding and I'm always the one that stays with [my chronically ill daughter]; that actually stressed me out probably nearly more than anything.*" (ID 09, child with end stage kidney failure). Though a sibling exemption for breast-feeding infants was granted, allowing infants to accompany the mother in the hospital, infants were not allowed in the intensive care unit, causing additional stress and costs. Managing outpatient appointments was similarly reported as challenging for families with young children, from both a practical and financial perspective.

## The impact of COVID-19 on support for families of children with chronic illness

Families of children with a chronic illness reported that they typically sought support from extended family, however risks and restrictions during the pandemic meant that this was often not possible. Community organizations were also reported as a vital support network for families. Unfortunately, many were limited in their capacity during the pandemic. The loss of charity funded accommodation, as well as social and psychological support, was keenly felt. Social support was reported as integral to well-being, with parents discussing how their local community, comprised of families of children without a chronic illness, suddenly became more understanding of what it was like to live with ongoing uncertainty and fear for their child's health. However, the valuable social support shared between families of children with a similar illness on the hospital wards was interrupted as many families tried to maintain social distancing and kitchens/communal areas were closed.

## COVID-19 interruptions to education

Home-based-learning was seen as highly challenging during periods of school closures, with one mother stating, "*I was literally ready to go 'See you all later, I am never coming back'. It was that bad.*"(ID 12, child with Dextrocardia situs inversus). Families of children with a chronic illness also raised unique issues, such as worries that if they ceased home schooling and returned their child to school, the spread of COVID-19 through their child's school might delay their child's treatment or surgery. Some parents shared that they also kept siblings home

even after schools were re-opened, potentially compromising their learning outcomes, due to fears that "catching something" would compromise the treatment protocols of their ill child.

Parents expressed that they felt the COVID-safe arrangements at the school, including not allowing parents on campus, meant that they were less able to communicate with teachers, resulting in missed feedback regarding their child's educational needs, and medical symptoms. Parents shared that they felt guilty and reluctant to send their child with additional needs to school during "closures", as schools had limited staff on campus.

Though parents often waited additional weeks before allowing their child to return to school after periods of closure, once most children had returned to school, the school stopped supporting home-based-learning leading to some parents feeling that their child was disadvantaged. However, other parents noted that the pandemic had actually helped their child who would not have been able to attend school due to illness, by increasing the availability of online learning platforms and resources. Some parents of chronically ill children also reported that they felt like they were better equipped to manage distance education than families of healthy children, due to past experiences.

Parents shared that they appreciated that schools and students were maintaining better cleaning and hygiene practices, and that children who were unwell or displaying any symptoms were discouraged from attending school. Parents noted that given these measures of prevention, their child had experienced fewer infections over the past several months with a notable positive impact on their health.

## Discussion

Despite experiencing lower levels of COVID-19 than many other parts of the world, especially during 2020, disruptions to health, education and social support have had an acute psychosocial impact on Australian families of children with chronic illness. These themes of mental health impact, feelings of loneliness and isolation, and changes to organisations and policies shaped by COVID-19, have been similarly identified among other populations where COVID-19 was more prevalent and school closures more prolonged [20]. Adequately responding to and addressing the needs of families of children with chronic illness, as well as planning for the minimization of longer-term impacts om well-being is critical.

As hospitals rapidly reorganized in response to the developing pandemic, many clinics were temporarily closed and clinicians were challenged to find alternative ways to continue to provide treatment and care while maintaining social distancing practices. For most, embracing telehealth was the obvious solution and the rapid uptake of virtual care was supported by new videoconferencing platforms that patients and their families found easy to access and use [21]. Positive attitudes towards pediatric telehealth [22, 23] are supported by our current study findings, with telehealth perceived to be convenient and one of the pandemic's "silver linings", which families hoped would remain after COVID-19 had abated. New models of care that integrate telehealth need to be developed, superseding the current model of repeatedly delivering stand-alone instances of use (i.e. the stop-gap use model) [24, 25]. Comprehensive models that offer a number of services, increase in complexity as needed, and address needs from diagnosis to disease progression and long-term care are required. Such programs are beginning to emerge, including Contactless [26], which specifically addresses the care needs of pediatric patients with rare and chronic conditions, who typically need regular follow-up even in the absence of acute events. Contactless is a multimodal, multidisciplinary remote-care model with a modular framework allowing basic care, with escalation on need, up to the more accurate and complex levels of care [26].

While telehealth has the potential to offer convenient care, improve access for rural families, and increase equity by decreasing the costs associated with accessing care for patients, more comprehensive evidence is needed to demonstrate that the telehealth model of care is at least equivalent to face to face care in terms of health outcomes. Also, telehealth may not be appropriate for families closer to diagnosis, whose child's management is still new, difficult, unfamiliar or fluctuating, and for families who are yet to establish a good working relationship with their clinical team. Special consideration of these newer families' needs should be taken into account when designing care pathways that incorporate or rely on telehealth.

Sydney Children's Hospital Network has reported a 55% increase in pediatric mental health presentations during the COVID-19 period [27]. The findings of this study confirm that there is a significant unmet need for psychosocial support among families of children with chronic illness. While this need most likely predated the pandemic, it is likely to have become stronger as stressors compound, services are stretched, and waitlists lengthen. This need for psychosocial support is reiterated in the emerging literature [1, 28–31] and new scales, such as the 'COVID-19 Family Stressor Scale' have been developed to measure COVID-related disruption and psychological stress [32]. It is important to remember, as many countries re-establish unrestricted movement and confidence, that many families of children with chronic illness will continue to prioritize their child's health over their family's need to return to normal. Among a community already fatigued by multiple lockdowns and pandemic stress, these families may experience an extended period of stress, fear, restrictions and diminished social interaction. It is critical to acknowledge that families of children with chronic illness may have a disproportionately difficult experience relative to other groups. Ensuring that appropriate psychosocial support services are available to families of children with chronic illness is a critical part of our community's roadmap out of the COVID-19 pandemic.

Families of children with chronic illness reported hospital visitation restrictions as one of the most challenging aspects of the pandemic, both practically and emotionally. During the pandemic, pediatric wards have been challenged with unique considerations in regards to visitation policies. While early evidence indicates visitor restriction is effective in preventing transmission of viruses within hospital settings [33], there are substantial complexities to be considered on pediatric wards. These include the dependence of young children on their parents for nutrition (e.g. breast feeding newborns), accomplishing daily-living tasks (e.g. dressing), decision-making (including medico-legal decisions), patient and parent education, and emotional comfort during traumatic illness and procedures.

Internationally, approximately 94% of North American hospitals with a pediatric ward, changed their visitor screening policy due to COVID-19 [34]. Though limiting visitors to children's hospitals is consistent internationally, we call on researchers and clinicians to re-imagine infection control within a family-centered model of care [35] that will limit parent reported increases in stress, feelings of emotional isolation and decreased support, decreased medical communication, marital discord, sibling distress, financial stress, chronic sleep deprivation and career interruption. Additional psychosocial support to families on the wards should be provided until revised models of COVID-safe, family-centered care can be implemented.

## Limitations

Our findings are based on the experiences of mothers only, with no fathers choosing to participate. Wade et. al. have shown that mothers have generally experienced greater COVID stress/ disruption, distress, anxiety, and post-traumatic stress symptoms, compared to fathers. In the Wade et. al. study mothers' higher COVID stress/disruption independently predicted all

mental health outcomes, suggesting a stress accumulation model [36]. This is consistent with our finding that mothers reported one of the most difficult aspects of the pandemic was the "juggling act" that they had to perform. It is also possible that hospitals' one-parent rule acts as a protective factor among fathers who, in this way, are shielded from distressing scenes of their child being ill, difficult conversations with medical professionals, and are not burdened by multiple responsibilities. Further research is needed to explore the experiences of fathers specifically.

This study is also limited to English speaking participants, recruited from a single site, and the recruitment of a small sample. Though the sample size of this study is small, sample adequacy in qualitative investigation relates to the appropriateness of the sample composition and size [37]. By purposively selecting families across a range of different diseases, ages, times since diagnosis, distances from the hospital and personal circumstances, the composition of this sample provides richly-textured and wide-ranging information relevant to the COVID-19 experience. Recent research has shown greater efficiency of purposive sampling (and accordingly lower sample size needs) compared to random sampling, as used in qualitative studies [38]. Furthermore, data saturation was observed within this sample, with no new themes reported within the last two interviews.

## Conclusions

The pandemic has, and will likely continue to go through waves, as new variants develop. It is critical that we understand the challenges and needs of families with chronically ill children and prepare a health and education system that will meet these needs in the future. Key lessons learned during the COVID-19 era can also inform innovations that generalize beyond the pandemic, including the benefits of advanced telehealth for rural and unwell families, the importance of psychosocial and family-centered care.

## Supporting information

**S1 Table. The perceived impact of COVID-19—illustrative quotations of longer length.** (DOCX)

## Author Contributions

**Conceptualization:** Jordana McLoone, Claire E. Wakefield, Glenn M. Marshall, Kristine Pierce, Raghu Lingam.

**Data curation:** Jordana McLoone.

**Formal analysis:** Jordana McLoone.

**Funding acquisition:** Raghu Lingam.

**Methodology:** Jordana McLoone, Raghu Lingam.

**Project administration:** Jordana McLoone.

**Resources:** Adam Jaffe, Ann Bye, Sean E. Kennedy, Donna Drew.

**Supervision:** Claire E. Wakefield, Glenn M. Marshall, Kristine Pierce, Raghu Lingam.

**Writing – original draft:** Jordana McLoone.

**Writing – review & editing:** Claire E. Wakefield, Glenn M. Marshall, Kristine Pierce, Adam Jaffe, Ann Bye, Sean E. Kennedy, Donna Drew, Raghu Lingam.

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
