## [Decision Letter · Decision Letter 0]

10 Jun 2022

PONE-D-22-14320It’s made a really hard situation even more difficult: The impact of COVID-19 on families of children with chronic illnessPLOS ONE

Dear Dr. McLoone,

Thank you for submitting your manuscript to PLOS ONE. After careful consideration, we feel that it has merit but does not fully meet PLOS ONE’s publication criteria as it currently stands. Therefore, we invite you to submit a revised version of the manuscript that addresses the points raised during the review process.

We look forward to receiving your revised manuscript.

Kind regards,

Stephane Shepherd, Ph.D

Academic Editor

PLOS ONE

Journal Requirements:

"The authors acknowledge the support of the Maridulu Budyari Gumal, The Sydney Partnership for Health, Education, Research & Enterprise (SPHERE) Child Unlimited Clinical Academic Group; CEW is supported by the National Health and Medical Research Council of Australia (APP2008300)."

Additional Editor Comments:

Thank you for your submission to PLOS ONE. The paper has now been reviewed by two experts in the field and I have also reviewed the paper.

Both reviewers recommended minor revisions. As such I invite you to respond to reviewer suggestions and re-submit your manuscript.

Reviewers' comments:

Reviewer's Responses to Questions

**Comments to the Author**

1. Is the manuscript technically sound, and do the data support the conclusions?

Reviewer #1: Yes

Reviewer #2: Yes

2. Has the statistical analysis been performed appropriately and rigorously? 

Reviewer #1: Yes

Reviewer #2: N/A

3. Have the authors made all data underlying the findings in their manuscript fully available?

Reviewer #1: Yes

Reviewer #2: No

4. Is the manuscript presented in an intelligible fashion and written in standard English?

Reviewer #1: Yes

Reviewer #2: Yes

5. Review Comments to the Author

Reviewer #1: Strengths:

• This study characterizes the pandemic experiences and challenges of families of children with chronic illness, identifying the need for additional supports and value of telehealth for this high risk population in such circumstances. It provides a unique perspective of parents of children with chronic illness, who are often overlooked.

• Several strategies that are helpful for management of children with chronic care needs were identified that can be applied in future pandemic situations as well as for ongoing routine care.

• Qualitative methodology allowed for rich data collection. Saturation was achieved in identification of common themes.

• Well written and compelling paper.

Weaknesses:

• Relatively small sample size of 13 parents, although saturation was reached in exploring themes.

• Heterogeneity of chronic illnesses where different challenges may have occurred.

• Did not include the perspective of children with chronic illness or health care providers, as has been reported in other similar studies.

Comments/Suggestions:

• This study provides a unique context, during the pandemic in Australia, whose incidence of COVID infection and public health measures differed from other countries. Other work exploring this issue in other countries occurred in a context where COVID was more prevalent and school closures were more prolonged, yet the issues identified are similar. (In addition to references cited, see also Nicholas et al, Pediatrics and Child Health, May 2022, https://doi.org/10. 1093/pch/pxab103). This is worth adding to the discussion, particularly to highlight what is unique about this study population compared to others reported in the literature.

• A more detailed explanation of the “Contactless” program alluded to in the discussion would be useful to the reader.

Reviewer #2: Thank you for the opportunity to review your paper which I read with great interest.

This is a robust qualitative paper that requires very little work to make it a paper that I think should be published.

There are some minor issues with punctuation that need to be addressed (see my comments in the attached manuscript re use of full stops with quotations). There are a few instances where an apostrophe is missing in it's.

I think Table 2 should be supplementary material rather than part of the main paper - it's a big table and the quality and comprehensiveness of the quotes already integrated into the text is already sound.

It would have been helpful if the COREQ checklist had been available for cross checking but this may not be a requirement of PLOS ONe.

The figure showing the complexity of overlap with the themes is good but the quality of the image is not great..... text looks fuzzy, so maybe this could be addressed.

Good luck with the paper and I look forward to seeing it published.

6. PLOS authors have the option to publish the peer review history of their article (what does this mean?). If published, this will include your full peer review and any attached files.

Reviewer #1: No

Reviewer #2: No

---

## [Author Response · Author response to Decision Letter 0]

4 Aug 2022

Dear Dr Sheperd and Reviewers,

Manuscript ID: PONE-D-22-14320

Title: It’s made a really hard situation even more difficult: The impact of COVID-19 on families of children with chronic illness

We thank the reviewers for taking the time to consdier the above-named manuscript submitted to PLOS One, and for their thoughtful suggestions in the email dated 11th June, 2022. We are grateful for the opportunity to make the suggested amendments and further enhance our manuscript. Please see below for our detailed responses to each of the reviewer’s comments (in bold) and find attached the revised manuscript for your consideration.

Editor’s comments:

We have reviewed all our files and are confident that they now meet the style requirements of the journal, including file naming. 

"The authors acknowledge the support of the Maridulu Budyari Gumal, The Sydney Partnership for Health, Education, Research & Enterprise (SPHERE) Child Unlimited Clinical Academic Group; CEW is supported by the National Health and Medical Research Council of Australia (APP2008300)."

Our sincere apologies for this confusion. We have deleted all funding information from the manuscript. In terms of updating the Funding Statement, we would like to acknowledge SPHERE. “The authors acknowledge the support of the Maridulu Budyari Gumal, The Sydney Partnership for Health, Education, Research & Enterprise (SPHERE) Child Unlimited Clinical Academic Group.” Thank you very much for amending this on our behalf. 

3. In your Data Availability statement, you have not specified where the minimal data set underlying the results described in your manuscript can be found. We will update your Data Availability statement to reflect the information you provide in your cover letter.

Apologies for not previously providing a Data Availability statement. Given that this research is qualitative in nature, the raw data (i.e. interview transcripts), contain highly sensitive and personal information. This information is medical and personal and refers to children (minors). There is no possible way of deidentifying the data in full due to countless referrals to identifiable information such as the participant’s name, the child’s name, demographic information, the child’s rare disease, the family’s unique situation, the time of year certain events occurred, the school, the hospital, etc. In addition, we are restricted by the relevant ethics committee to not allow access to this data unless expressly named in the approved application. All requests for access to data can be made to either myself, Jordana Mcloone (J.Mcloone@unsw.edu.au) or to the ethics committee, quoting the study reference number: 2020/ETH02434. 

The Sydney Children’s Hospital Network Human Research Ethics Committee (SCHN HREC)

Email: SCHN-Ethics@health.nsw.gov.au

Phone +612 9845 1253. 

Mail: ‘Research Ethics’ at the Sydney Children’s Hospital Network, Locked Bag 4001, Westmead, 2145, NSW, Australia

The reference list has been checked and we are confident all papers are cited correctly.

Reviewer #1: 

We would like to thank Reviewer 1 for their kind comments and acknowledgement of the study’s strengths. In response to their “Comments/Suggestions”, we would like to share how we have amended the manuscript to improve upon these aspects.

1. This study provides a unique context, during the pandemic in Australia, whose incidence of COVID infection and public health measures differed from other countries. Other work exploring this issue in other countries occurred in a context where COVID was more prevalent and school closures were more prolonged, yet the issues identified are similar. (In addition to references cited, see also Nicholas et al, Pediatrics and Child Health, May 2022, https://doi.org/10. 1093/pch/pxab103). This is worth adding to the discussion, particularly to highlight what is unique about this study population compared to others reported in the literature.

We thank you for bringing this paper to our attention and have included this reference in our manuscript, as well as alluding to your greater point about the similarity of themes, despite the differences in context. 

“Despite experiencing lower levels of COVID-19 than many other parts of the world, especially during 2020, disruptions to health, education and social support have had an acute psychosocial impact on Australian families of children with chronic illness. These themes of mental health impact, feelings of loneliness and isolation, and changes to organisations and policies shaped by COVID-19, have been similarly identified among other populations where COVID-19 was more prevalent and school closures more prolonged (1) Adequately responding to and addressing the needs of families of children with chronic illness, as well as planning for the minimization of longer-term impacts om well-being is critical.” 

2. A more detailed explanation of the “Contactless” program alluded to in the discussion would be useful to the reader.

Thank you for this helpful suggestion. We have added some additional information about Contactless, which does not do such a complex project justice, but will at least allow readers to appreciate its model as they read. 

“Comprehensive models that offer a number of services, increase in complexity as needed, and address needs from diagnosis to disease progression and long-term care are required. Such programs are beginning to emerge, including Contactless (25), which specifically addresses the care needs of pediatric patients with rare and chronic conditions, who typically need regular follow-up even in the absence of acute events. Contactless is a multimodal, multidisciplinary remote-care model with a modular framework allowing basic care, with escalation on need, up to the more accurate and complex levels of care (2).”

Reviewer #2: 

We would also like to thank Reviewer 2 for their insightful comments and suggestions. 

1. There are some minor issues with punctuation that need to be addressed (see my comments in the attached manuscript re use of full stops with quotations). There are a few instances where an apostrophe is missing in it's.

We sincerely thank Reviewer 2 for taking the time to make such detailed edits on the manuscript. We have transferred and amended each of these edit notes to the track-changed version and clean version of the revised manuscript, and ran a search of all the it/it’s throughout the manuscript paying close attention to ensure we captured and revised each instance. 

2. I think Table 2 should be supplementary material rather than part of the main paper - it's a big table and the quality and comprehensiveness of the quotes already integrated into the text is already sound.

We agree and have confirmed with the journal that this is their preference also. Table 2 is now included as Supplementary material. 

3. It would have been helpful if the COREQ checklist had been available for cross checking but this may not be a requirement of PLOS ONe.

We have checked with the journal and they communicated that this is not a requirement as you suggested may be the case. However, we have made a reference to the COREQ in the methods section. 

“We used the COREQ (19) checklist to promote accurate reporting of qualitative studies.”

4. The figure showing the complexity of overlap with the themes is good but the quality of the image is not great..... text looks fuzzy, so maybe this could be addressed.

We have checked this thoroughly and used the journal’s image enhancer software to obtain the greatest quality possible for the image. We have also flagged this with the journal and asked that their formatting and production team work with the image to achieve the best possible clarity. 

Good luck with the paper and I look forward to seeing it published.

I hope you will agree that we have adequately addressed the queries raised by the reviewers. Should you have any further questions, please do not hesitate to contact.

Thank you for considering our revised manuscript. We look forward to your response.

Regards,

Jordana MdLoone

---

## [Editor Report · Decision Letter 1]

12 Aug 2022

It’s made a really hard situation even more difficult: The impact of COVID-19 on families of children with chronic illness

PONE-D-22-14320R1

Dear Dr. McLoone,

We’re pleased to inform you that your manuscript has been judged scientifically suitable for publication and will be formally accepted for publication once it meets all outstanding technical requirements.

Kind regards,

Stephane Shepherd, Ph.D

Academic Editor

PLOS ONE
---

## [Editor Report · Acceptance letter]

23 Aug 2022

PONE-D-22-14320R1 

*It’s made a really hard situation even more difficult*: The impact of COVID-19 on families of children with chronic illness 

Dear Dr. McLoone:

I'm pleased to inform you that your manuscript has been deemed suitable for publication in PLOS ONE. Congratulations! Your manuscript is now with our production department. 

Kind regards, 

on behalf of

Dr. Stephane Shepherd 

Academic Editor

PLOS ONE